# The breakdown of both strange metal and superconducting states at a pressure-induced quantum critical point in iron-pnictide superconductors

Shu Cai[1], Jinyu Zhao [1,2], Ni Ni [3,4], Jing Guo [1,5], Run Yang[1], Pengyu Wang[1,2], Jinyu Han[1,2], Sijin Long[1,2], Yazhou Zhou[1], Qi Wu[1], Xianggang Qiu[1,2,5], Tao Xiang [1,2], Robert J. Cava[3] & Liling Sun [1,2,5] ✉

Here we report the first observation of the concurrent breakdown of the strange metal (SM) normal state and superconductivity at a pressure-induced quantum critical point in $Ca_{10}(Pt_4As_8)((Fe_{0.97}Pt_{0.03})_2As_2)_5$ superconductor. We find that, upon suppressing the superconducting state, the power exponent ($\alpha$) changes from 1 to 2, and the slope of the temperature-linear resistivity per FeAs layer ($A^\square$) gradually diminishes. At a critical pressure, $A^\square$ and superconducting transition temperature ($T_c$) go to zero concurrently, where a quantum phase transition from a superconducting state with a SM normal state to a non-superconducting Fermi liquid state occurs. Scaling analysis reveals that the change of $A^\square$ with $T_c$ obeys the relation of $T_c \sim (A^\square)^{0.5}$, similar to what is seen in other chemically doped unconventional superconductors. These results suggest that there is a simple but powerful organizational principle of connecting the SM normal state with the high-$T_c$ superconductivity.

The strange metal (SM) state is an extraordinary normal state of high-temperature superconductors in which the electrical resistivity grows linearly with temperature in the low-temperature limit. Such a unique state has been found in the doped cuprates, iron-pnictide superconductors[1–10], organic materials[8], heavy Fermion metals[11–13], infinite-layer nickelates[14] and twisted bilayer graphene[15]. Although many efforts have been made in past decades, the correlation of the SM state with the superconductivity still remains a subject of debate, which requires more experimental information from different experimental probes and materials. The central questions for this key issue are what is the intrinsic correlation between the SM normal state and the superconductivity, and what is the determining factor associated with the SM normal state for stabilizing high-$T_c$ superconductivity. To find clues for solving these puzzles, a superconductor with a pure SM normal state featured by $\alpha = 1$ (in the form of $\rho = \rho_0 + AT^\alpha$) is needed.

And better, if a superconducting system only hosts as few as possible competing orders in the tuning parameter−temperature phase diagram, then direct information to reveal the relationship between these two may be obtained.

The iron-pnictide superconductor $Ca_{10}(Pt_4As_8)((Fe_{0.97}Pt_{0.03})_2As_2)_5$ is such an ideal superconducting material for this kind of study. The ground state of this superconductor is well understood, without the complication from ordered states, such as pseudogap state or a nearby long-range AFM state, etc[16]. More intriguingly, it shows a $T$-linear resistivity behavior in its normal state at ambient pressure[16], providing us a unique platform for studying the correlation between the $T$-linear resistivity and the superconductivity.

Pressure is one of the non-thermal control parameters for tuning superconductivity because this method can shorten the interatomic distances and correspondingly leads to the change of the crystal and

[1]Institute of Physics, Chinese Academy of Sciences, 100190 Beijing, China. [2]University of Chinese Academy of Sciences, 100190 Beijing, China. [3]Department of Chemistry, Princeton University, Princeton, New Jersey 08544, USA. [4]Department of Physics and Astronomy, UCLA, Los Angeles, CA 90095, USA. [5]Songshan Lake Materials Laboratory, 523808 Dongguan, Guangdong, China. ✉e-mail: llsun@iphy.ac.cn

electronic structures without altering the chemistry. Moreover, it can avoid system uncertainties associated with the differences between the samples with different doping levels, which poses significant limitations on the studies of chemically doped single crystals. Thus, high pressure, as a 'clean' way, has been widely adopted as an independent control parameter to explore new phenomena and the evolution from one state to another in correlated electron systems[17–25]. In this study, we take the $Ca_{10}(Pt_4As_8)((Fe_{0.97}Pt_{0.03})_2As_2)_5$ superconductor (here referred to as the "1048 superconductor") as a target material to study the pressure-induced co-evolution of the SM and superconducting states.

## Results and discussion
### Structure information and transport properties
The $Ca_{10}(Pt_4As_8)(Fe_2As_2)_5$ superconductor crystallizes in a tetragonal unit cell with -Ca-$(Pt_4As_8)$-Ca-$(Fe_2As_2)$-stacking[16]. Its structure can be described as a distorted $CaFe_2As_2$ structure with every other $Fe_2As_2$

layer replaced by a $Pt_4As_8$ layer, as shown in Fig. 1a. At ambient pressure, its resistivity versus temperature shows a linear behavior above the superconducting transition at 26 K (Fig. 1b), which closely resembles what is seen in optimally doped cuprate superconductors[2,26,27]. When pressure is applied on the samples (S#1 and S#2) surrounded by the pressure transmitting medium (PTM) of NaCl, we find that $T_c$ shifts to low temperature upon compression and is invisible at 13 GPa and above for the sample #1 (Fig. 1c, d), indicating that the application of pressure can suppress the superconductivity effectively. We repeat the measurements with new samples cut from different batches and obtain reproducible results (Fig. 1e, f)—increasing pressure brings a monotonic reduction in $T_c$, and the superconductivity vanishes for the S#2 at 11.8 GPa. These results demonstrate the reproducible evolution of the superconductivity of the 1048 superconductor with pressure.

To clarify the possibility that the suppression of $T_c$ is related to the pressure inhomogeneity introduced by the solid pressure medium NaCl, we perform the high-pressure measurements on the S#4 with the

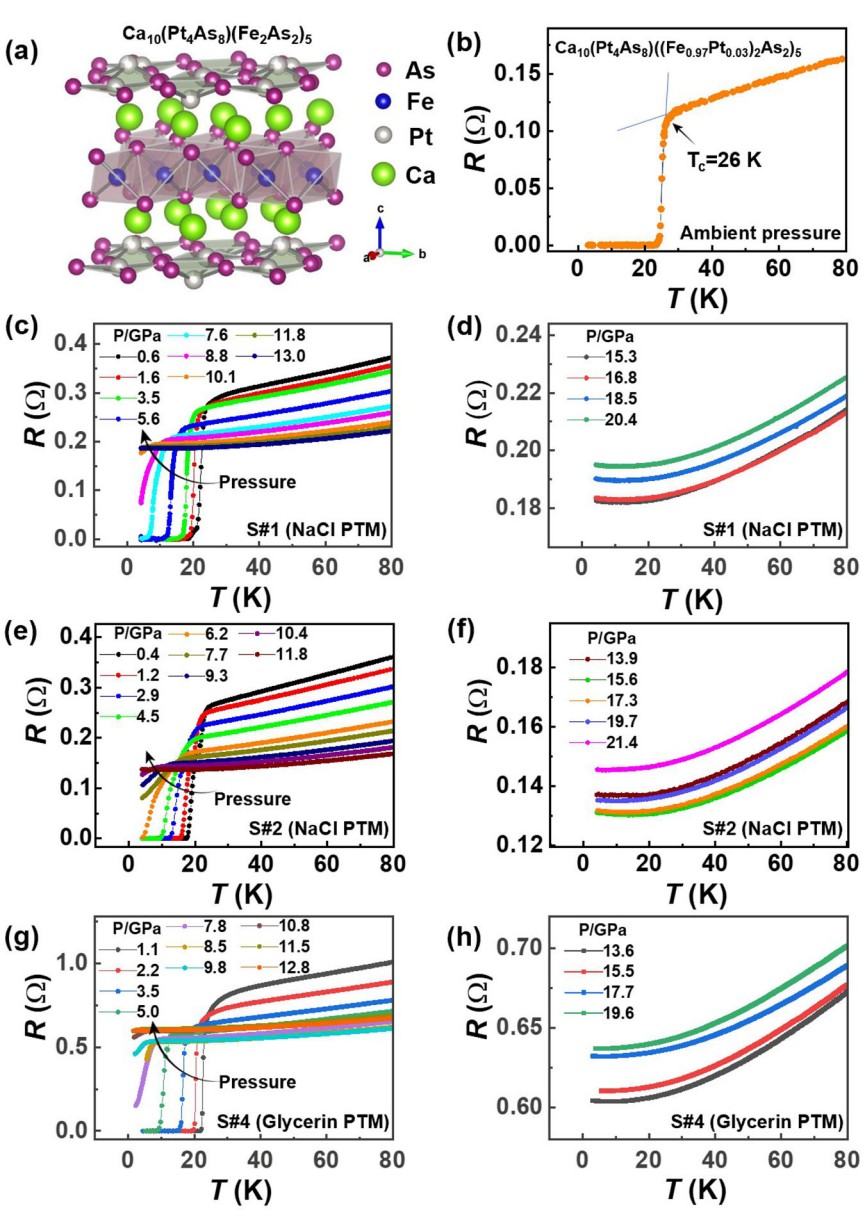

**Fig. 1 | Structure information and transport properties for the 1048 superconductor. a** The crystal structure of the 1048 superconductor. **b** Ambient-pressure resistance as a function of temperature, displaying a superconducting transition with an extradentary sharp drop at 26 K. **c**–**f** Temperature dependence of resistance for the S#1 and S#2 surrounded by the pressure transmitting medium (PTM) of NaCl in the measurements. **g**, **h** Resistance versus temperature for the S#4 surrounded by the liquid PTM of glycerin in the measurements.

liquid PTM of glycerin, and find the same results (Fig. 1g, h)−$T_c$ declines with increasing pressure and vanishes at 12.8 GPa. Moreover, it is seen that the superconducting transition of the sample in the liquid pressure environment appears to have a sharper drop and no low-temperature tail at the foot of the transition at low pressures up to 4 GPa (Fig. 1c, e and g). This indicates that the different PTMs have an appreciable effect on superconductivity.

**High-pressure resistance versus temperature and corresponding fits**
We plot the resistance versus temperature for the pressurized S#1 and the S#4 (Fig. 2) and make the actual fits to the temperature ($T$) dependence of the normal state resistance at low temperatures for the data obtained from the compressed samples based on the following equations:

$$R(T) = R'_0 + A'T + B'T^2$$

$$R(T) = R_0 + AT^\alpha$$

Where $R_0$ and $R'_0$ are the residual resistance, $A'$, $B'$, $A$ and $\alpha$ are the slope of the $T$-linear resistance, the coefficient of $T$-square resistance, the coefficient of power function and the power exponent of temperature, respectively.

Since the determination of the resistivity as a function of temperature at each pressure investigated requires the pressure dependence of sample thickness, which is difficult to be obtained in the high-pressure resistance measurements, we replace resistivity with resistance in the data fitting so as to take no account of the relation between the applied pressure and the sample thickness (see "Methods"). It has been known that high-temperature superconductors such as cuprates and iron pnictides have a layered crystal structure; their superconductivities display prominent two-dimensional characteristics[28–31]. To describe the correlation between the SM state and superconductivity reasonably, we employ $A^\square$, the $T$-linear resistivity coefficient normalized by the average distance between FeAs layers, to investigate this important correlation (see "Methods"). The fitting is in good consistence with the experiment results (see red lines in Fig. 2), giving a set of coefficient $A^\square$ that varies with pressure from 10.8 at 0.6 GPa to 0 at -13 GPa and above for the S#1 (Fig. 2a–h), and from 12.8 at 1.1 GPa to 0 at -12.8 GPa and above for the S#4 (Fig. 2i–p). The other fitting results can be found in Supplementary Information (SI). We extract the power exponent ($\alpha$), based on the equation in the form of power law, as a function of pressure and find that pressure changes $\alpha$ from 1 at ambient pressure to 2 at 13 GPa for the S#1 and 12.8 GPa for the S#4. When pressure is higher than -13 GPa for S#1 and 12.8 GPa for S#4, the low-temperature resistance shows curvature (Fig. 2e–h, m–p) and approaches a $T^2$ dependence in the higher pressure range, where the superconductivity is fully suppressed. These results indicate that the application of pressure gradually drives the 1048 superconductor from a superconducting ground state with a SM normal state to a non-superconducting Fermi-liquid (FL) state.

**$T_c$, $\alpha$ and $A^\square$ versus pressure**
We summarize the experimental results in the pressure–temperature phase diagram in Fig. 3a. There are three distinct regions in the diagram: the superconducting (SC) state, the strange metal (SM) state (or non-Fermi liquid (NFL) behavior) and the non-superconducting Fermi liquid (FL) state. It is seen that the ambient-pressure superconductivity of the sample develops from a pure SM normal state, featured by $\alpha = 1$ (Fig. 3b), and the sample holds the highest $T_c$ value. Upon increasing pressure till the critical pressure $P_c$, $T_c$ decreases monotonously, with $\alpha$ varying between 1 and 2 (here, we take the average value of the pressures that destroy the superconducting state, obtained from different

experimental runs, as $P_c$, see Fig. 3). At $P_c$ (-12.5 GPa), the superconductivity is completely suppressed (Fig. 3a) and the power exponent $\alpha$ approaches 2 (Fig. 3b), indicating that the ground state of the sample moves into a non-superconducting FL state. These results demonstrate that applying pressure renders the 1048 superconductor to undergo a quantum phase transition at -12.5 GPa. Below the quantum critical point (QCP), the superconductor has a SC ground state with a SM normal state or NFL behavior, while above the QCP, the sample displays a non-superconducting FL state. The pressure-induced quantum phase transition observed in the 1048 superconductor is highly reminiscent of what has been seen in the over-doped cuprate superconductors[2,3,5,6,26] and Fe-based superconductor[32,33], which show the same evolution upon increasing doping level and gating voltage.

We track the change of the coefficient $A^\square$ as a function of pressure before reaching the QCP. As shown in Fig. 3c, the coefficient $A^\square$ versus pressure displays a similar trend with $T_c$ versus pressure. As $A^\square$ holds the maximum, the 1048 superconductor displays the highest $T_c$ value. Once $A^\square$ is decreased by pressure, $T_c$ exhibits a decrease correspondingly. More significantly, we find that the values of the coefficient $A^\square$ and $T_c$ reach zero together at the QCP. The observation of the synchronized decrease of $A^\square$ and $T_c$ with applied pressure leads us to propose that the coefficient $A^\square$ is a key factor for determining the $T_c$ value of the high-$T_c$ superconductors.

**Relationship between superconductivity and strange metal**
To establish the correlation between the SM normal state and $T_c$ for the 1048 superconductor and compare it with doped cuprate and other superconductors, we plot the $T_c$ dependence of the coefficient $A^\square$ in Fig. 4a and find a positive correlation of $A^\square$ with $T_c$. Scaling analysis on the two quantities finds that $T_c$ versus $A^\square$ basically obeys the relation of $T_c \sim (A^\square)^{0.5}$ (see the inset of Fig. 4a). To the best of our knowledge, this is the first quantitative description of the correlation between $T_c$ and $A^\square$ for the iron-pnictide superconductors through the pressure tuning method. For the sake of investigating the possible universality of the concurrent breakdown of the SM normal state and superconducting state, we have also performed hydrostatic pressure studies on $Sr_{0.74}Na_{0.26}Fe_2As_2$ superconductor, whose normal resistance versus temperature displays a FL state at ambient pressure but turns into a SM state at 5.9 GPa (see Supplementary Fig. S3). The results obtained also show the concurrent breakdowns of $A^\square$ and $T_c$ at the same pressure point (see Supplementary Fig. S3c). Further, we compare our results with the experimental data obtained from chemically doped cuprate[6,26,34] and iron-based superconductors[8,35,36] and find a similar relationship between $T_c$ and $A^\square$ (Fig. 4b), demonstrating that the stability of the superconducting phase has an intimate connection with the contribution from the electron state of determining the $T$-linear resistivity.

It has been proposed that antiferromagnetic (AFM) spin fluctuations are responsible for both the superconducting and SM normal states in the high-$T_c$ superconductors[34,37]. If the AFM fluctuations are the fundamental principle, then a theory to explain the detailed physics related to the behavior of the SM state, as revealed in this study, is also needed.

In conclusion, our results reveal that the application of pressure induces a quantum phase transition (QPT) from a SC ground state with a SM normal state to a non-superconducting FL state in the $Ca_{10}(Pt_4As_8)((Fe_{0.97}Pt_{0.03})_2As_2)_5$ superconductor. The observed pressure-tuned co-evolution of the strange metal (SM) and superconducting (SC) states, as well as the concurrent breakdown of the SM and SC states at the quantum critical pressure (-12.5 GPa), demonstrates how the coefficient $A^\square$, the slope of the linear-in-temperature resistivity normalized by the average distance between FeAs layers, determines the stability of the superconductivity and the value of $T_c$. Below the QCP, the coefficient $A^\square$ is continuously suppressed from a finite value, and the power component $\alpha$ varies between 1 and 2. Above

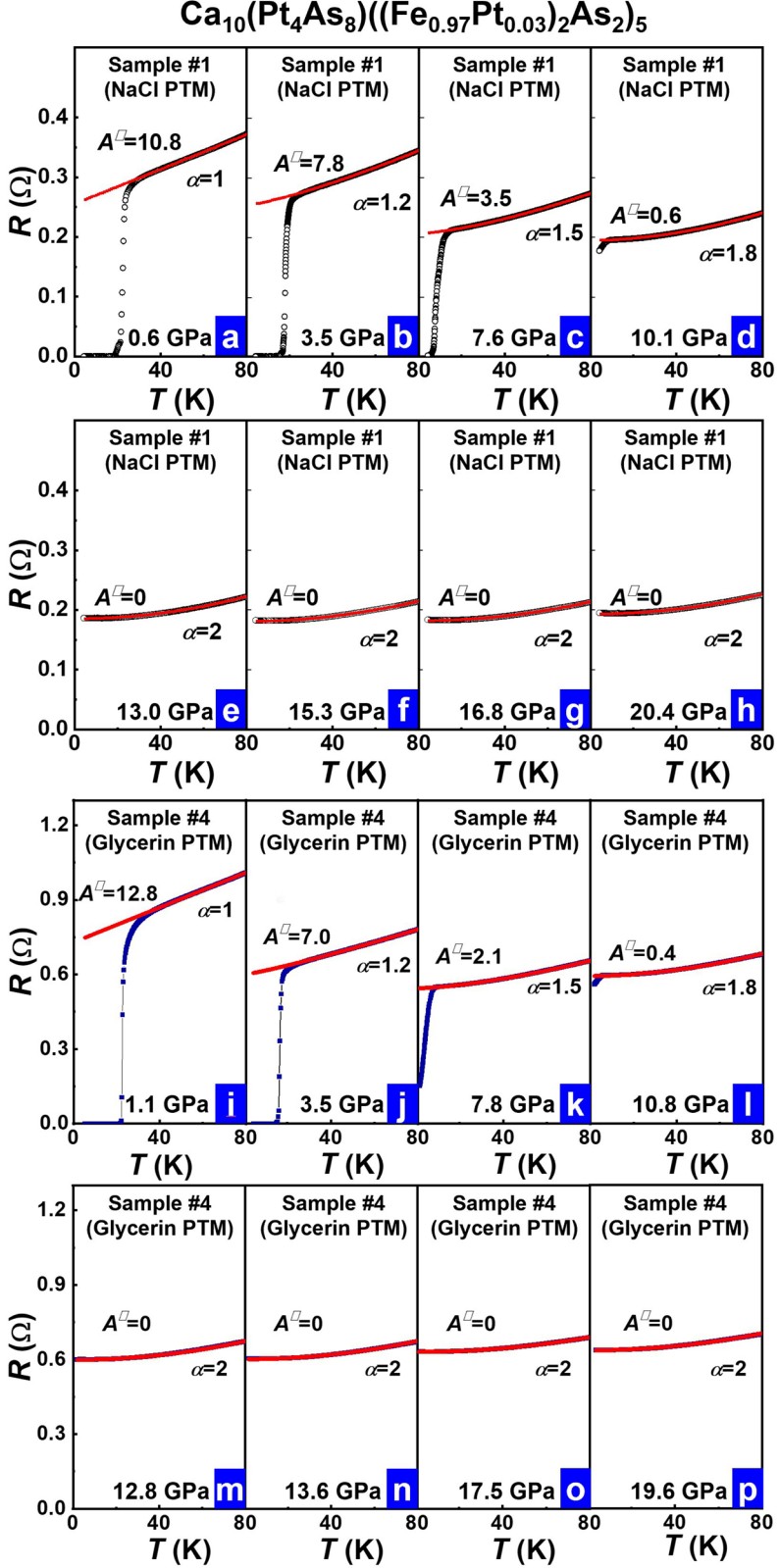

$$Ca_{10}(Pt_4As_8)((Fe_{0.97}Pt_{0.03})_2As_2)_5$$

the QCP, the SC state is fully suppressed, while $A^{\square}=0$ and $\alpha=2$ simultaneously, a hallmark of a pure Fermi liquid (FL) state. We have also observed the same phenomenon in pressurized $Sr_{0.74}Na_{0.26}Fe_2As_2$ superconductor. The scaling analysis for the obtained $T_c$ and coefficient $A^{\square}$ in the compressed 1048 superconductor finds that it basically obeys the relation of $T_c \sim (A^{\square})^{0.5}$, exhibiting the similarity with what is seen in other chemically doped unconventional superconductors and demonstrating for the first time that the high-$T_c$ superconductors with a SM normal state obey the same relation in $T_c(A^{\square})$, regardless of the type of the tuning method (doping or pressurizing), the crystal structure, the bulk or film superconductors and the nature of dopant (electrons or holes). The analysis of the generic relation between the $T_c$ and the SM state for the cuprate and iron-based high-$T_c$ superconductors reveals that the same physics

**Fig. 2 | High-pressure resistance as a function of temperature at different pressures for the Ca$_{10}$(Pt$_4$As$_8$)((Fe$_{0.97}$Pt$_{0.03}$)$_2$As$_2$)$_5$ superconductors and corresponding fits by the forms of $R = R'_0 + A'T + B'T^2$. a** $A^\square$ ($A^\square = (w/l)(t/d)A'$, here $w$ and $t$ are the width and the thickness of the sample, $l$ is the distance between the electrodes, and $d$ is an average distance between FeAs layers (see "Methods") shows the maximum value (10.8) at 0.6 GPa for the S#1 surrounded by the pressure transmitting medium (PTM) of NaCl, where the normal state of the sample is in a pure SM state ($\alpha = 1$), and its $T_c$ displays the highest value. **b–d** $A^\square$ decreases with applied pressure in the pressure range of 3.5 GPa–10.1 GPa for the S#1, in the

pressure range of which $T_c$ is suppressed gradually. **e–h** $A^\square$ approaches zero and $\alpha = 2$ for the S#1 in the pressure range of 13.0 GPa–20.4 GPa, in which the superconductivity is absent. **i** $A^\square = 12.8$ at 1.1 GPa for the S#4 surrounded by the PTM of glycerin, where the normal state of the sample is also in a pure SM state ($\alpha = 1$), and its $T_c$ displays the highest value. **j–l** $A^\square$ declines upon increasing pressure for the S#4 in the pressure range of 3.5 GPa–10.8 GPa. **m–p** $A^\square = 0$ and $\alpha = 2$ for the S#4 in the pressure range of 12.8 GPa–19.6 GPa, in which the sample is no longer superconducting.

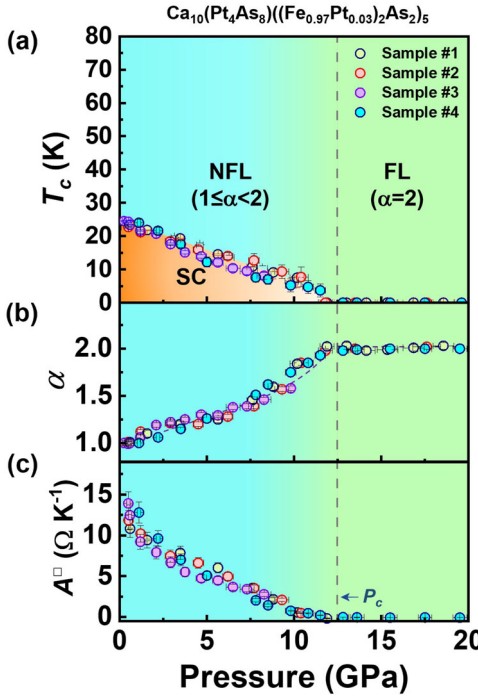

**Fig. 3 | Summary of the superconductivity, power exponent ($\alpha$) and the coefficient ($A^\square$) as a function of pressure for the Ca$_{10}$(Pt$_4$As$_8$)((Fe$_{0.97}$Pt$_{0.03}$)$_2$As$_2$)$_5$ superconductor. a** Pressure–$T_c$ phase diagram. NFL and FL represent the non-Fermi liquid behavior and Fermi liquid state, respectively. SC stands for the superconducting state. The $T_c$ value in the phase diagram is determined by the onset transition temperature. **b** The plot of $\alpha$ in the form of $R(T) = R_0 + AT^\alpha$ versus pressure. **c** Pressure dependence of the coefficient $A^\square$. The error bars represent the s.d.

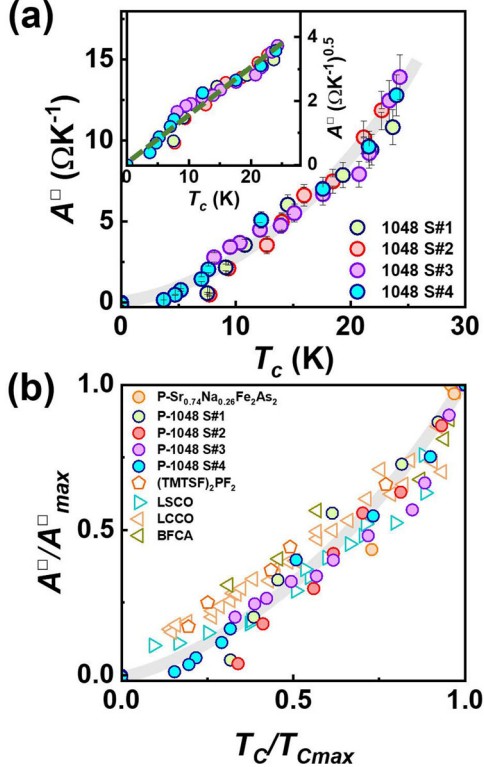

**Fig. 4 | Superconducting transition temperature versus slope of linear-in-temperature resistivity for the pressurized Ca$_{10}$(Pt$_4$As$_8$)((Fe$_{0.97}$Pt$_{0.03}$)$_2$As$_2$)$_5$ superconductor. a** The coefficient $A^\square$ as a function of superconducting transition temperature ($T_c$). The error bars represent the s.d. The inset displays $T_c$ dependence of $(A^\square)^{0.5}$. **b** The summary of the correlation between $A^\square/A^{\square max}$ and $T_c/T_c^{max}$ for the pressurized Ca$_{10}$(Pt$_4$As$_8$)((Fe$_{0.97}$Pt$_{0.03}$)$_2$As$_2$)$_5$ and Sr$_{0.74}$Na$_{0.26}$Fe$_2$As$_2$ superconductors, as well as other differently doped superconductors. The LCCO, LSCO and BFCA represent La$_{2-x}$Ce$_x$CuO$_4$ superconductor [ref. 26], La$_{2-x}$Sr$_x$CuO$_4$ superconductor [refs. 6,26] and Ba(Fe$_{1-x}$Co$_x$)$_2$As$_2$ superconductor [refs. 26,35,36], respectively. The data about (TMTSF)$_2$PF$_2$ are taken from ref. 8.

governs the emergence and the stability of the high-$T_c$ superconductivity.

## Methods

### Single crystal growth

The high-quality single crystals of Ca$_{10}$(Pt$_4$As$_8$)(Fe$_2$As$_2$)$_5$ were grown by a solid-state reaction method. CaAs, FeAs, Fe, Pt, and As were mixed in an argon-filled glovebox, pressed into pellets, and sealed in quartz tubes under 1/3 atmosphere of Ar. The tubes were heated to 1100–1180 °C for 1 week, furnace-cooled to 900 °C for 1 day, and then water quenched. The pure10-4-8 phase single crystals with 0.5 × 0.5 × 0.03 mm$^3$ size were obtained. The high-quality single crystals of Sr$_{0.74}$Na$_{0.26}$Fe$_2$As$_2$ were synthesized using a self-flux technique. The detailed sample preparation procedure for these two materials can be found in refs. 16,38,39.

### Experimental details for high-pressure measurements

High pressure was generated by a diamond anvil cell made of BeCu alloy with two opposing anvils. A four-probe method was applied for our resistance measurements. Diamond anvils with 300 and 400 μm culets (flat area of the diamond anvil) were used for several

independent measurements. In the experiments, we employed platinum foil as electrodes, rhenium plate as gasket and cubic boron nitride as insulating material. Two kinds of pressure-transmitting media, NaCl and glycerin, were used for the different runs of high-pressure measurements. Pressure in all measurements is determined by the ruby fluorescence method at room temperature[40] and then calibrated by pressure dependence of Pb's superconducting transition temperature in the same pressure cell.

The sizes of the sample #1, sample #2 and sample #4 are 100 × 60 × 5 μm$^3$, 90 × 60 × 5 μm$^3$ and 110 × 40 × 3 μm$^3$, respectively.

### Extraction method for the coefficient $A^\square$

Usually, the coefficient $A$ can be extracted by fitting the plot of resistivity $\rho$ versus temperature $(T)$ in the form of $\rho(T) = \rho_0 + AT + BT^2$, where $\rho_0$, $A$, $B$ are the residual resistivity, the slope of $T$-linear resistivity, the

slope of $T$-square resistivity, respectively. In practice, the resistivity $\rho$ is commonly derived by the formula: $R = \rho l/wt$, where $R$ is the measured resistance, $l$ is the distance between the electrodes for the voltage measurements, $w$ and $t$ are the width and the thickness of the sample. In our experiments, we use a method suitable for layered compounds to extract $A^{\square}$ by fitting the resistance $R$ in the form of $R(T) = R_0 + AT + BT^2$ instead of resistivity $\rho$ in the form of $\rho(T) = \rho_0 + AT + BT^2$. By adopting this method, we analyzed the pressure effect on the variation of the sample thickness that can be neglected.

We start with the following equation:

$$\rho(T) = \rho_0 + AT + BT^2 \tag{1}$$

If the two sides of the equation are multiplied by $l/wt$, the equation can be written as:

$$R(T) = (l/wt)\rho_0 + (l/wt)AT + (l/wt)BT^2 \tag{2}$$

When $A$ is expressed by $A^{\square}$ ($A = A^{\square}d$, $d$ is the average distance between FeAs layers), then the equation can be changed to:

$$R(T) = (l/wt)\rho_0 + (A^{\square}l/w)(d/t)T + (l/wt)BT^2 \tag{3}$$

Let us consider the case of $R'_0 = (l/wt)\rho_0$, $A' = (A^{\square}l/w)(d/t)$ and $B' = (l/wt)B$, the Eq. (3) can be expressed as:

$$R(T) = R'_0 + A'T + B'T^2 \tag{4}$$

In this study, $R(T)$ measured at different pressures are fitted by Eq. (4). Since $R'_0$, $A'$ and $B'$ are all the fitting parameters that can be obtained directly from the fitting results of $R(T)$, these values of $R'_0$, $A'$ and $B'$ are known. Then, we can use Eq. (5) to extract the pressure dependence of $A^{\square}$ for the 1048 superconductor:

$$A^{\square} = (w/l)(t/d)A' \tag{5}$$

Next, we analyze the pressure effect on the quantities in Eq. (5). Considering that the 1048 superconductor possesses the tetragonal structure[16]—its lattice parameter $a = b$, we propose that the width ($w$) and distance ($l$) between the electrodes for the voltage measurements of our ambient-pressure sample can be expressed as:

$$w_0 = Xa$$

$$l_0 = Ya$$

where $X$ and $Y$ are the numbers of the unit cells, and $a$ is the lattice parameter, respectively. The ratio of $w_0/l_0$ equals $X/Y$. Since $w$ and $l$ are the function of lattice parameter $a$, its ratio ($w_p/l_p$) at different pressures should be the same as that obtained at ambient pressure, i.e., $w_0/l_0 = w_p/l_p = X/Y$. This allows us to employ the ratio of $w_0/l_0$ for the determination of $A^{\square}$ as a function of pressure. The values of $w_0$ and $l_0$ can be found in the section "Experimental details for high-pressure measurements."

Since the bulk 1048 superconductor is composed of a number of unit cells, sample thickness $t$ can thus be expressed as $t = Nc$, where $c$ is the lattice parameter perpendicular to the FeAs layer in the unit cell, and $N$ is the number of the unit cells. It has been known that each unit cell of the 1048 superconductor contains only one FeAs layer (Fig. 1a); thus, the average distance between the FeAs layer, $d$, equals the lattice parameter $c$ ($d = c$). At ambient pressure, the ratio of the thickness $t_0$ and the

average distance between FeAs layer $d_0$ should be in the form of:

$$t_0/d_0 = N \tag{6}$$

Because both values of $t$ and $d$ are the function of lattice parameter $c$, when pressure is applied, the ratio of $t_p/d_p$ still equals $N$, i.e., $t_p/d_p = t_0/d_0 = N$. This allows us to use the ratio of $t_0/d_0$ to establish the pressure dependence of $t_p/d_p$. The $t_0/d_0$ can be found in our experimental details and ref. 16.

Based on the above analysis, through the fitting $R(T)$ measured at different pressures, the coefficient $A^{\square}$ can be extracted as:

$$A^{\square}(P) = (w_0/l_0)(t_0/d_0)A' \tag{7}$$

## Data availability
The data that support the findings of this study are available from the corresponding author upon request.

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

## Acknowledgements
This work in China was supported by the NSF of China (Grant Nos. U2032214, 12104487, 12122414 and 12004419), the National Key Research and Development Program of China (Grant No. 2021YFA1401800), and the Strategic Priority Research Program (B) of the Chinese Academy of Sciences (Grant No. XDB25000000). J.G. and S.C. are grateful for support from the Youth Innovation Promotion Association of the CAS (2019008) and the China Postdoctoral Science Foundation (E0BK111). The work at Princeton was supported by the US Department of Energy, Division of Basic Energy Sciences, grant DE-FG02-98ER-45706. Work at UCLA was supported by the U.S. Department of Energy (DOE), Office of Science, Office of Basic Energy Sciences under Award Number DE-SC0021117.

## Author contributions
L.S., T.X. and Q.W. designed the study and supervised the project. N.N. and R.J.C. grew the $Ca_{10}(Pt_4As_8)(Fe_2As_2)_5$ single crystals. R.Y. and X.G.Q. grew the $Sr_{0.74}Na_{0.26}Fe_2As_2$ single crystals. S.C., J.Y.Z., J.G., P.Y.W., J.Y.H., S.J.L., Y.Z.Z. and L.S. performed the high-pressure resistance measurements. L.S., T.X., Q.W., S.C. and R.J.C. wrote the manuscript in consultation with all authors.

## Competing interests
The authors declare no competing interests.
