## [Peer Review File · Nature Communications]

The breakdown of both strange metal and superconducting states at a pressure-induced quantum critical point in iron-pnictide superconductorsEditorial Note: This manuscript has been previously reviewed at another journal that is not operating a transparent peer review scheme. This document only contains reviewer comments and rebuttal letters for versions considered at *Nature Communications*.

Reviewer #1 (Remarks to the Author):

I have read carefully the revised manuscript by Cai et al., submitted to Nat. Commun., and the rebuttal letter in which the authors address the concerns and criticism of the referees of the previous review process. As far as I can tell, the replies are mostly to the point, and the authors have made sufficient changes for publishing the manuscript.

However, I would like to suggest out two points which might be worth addressing beforehand: Firstly, it is certainly impressive that the authors have repeated their measurements with a liquid PTM. Unfortunately, their choice for Glycerine was probably not ideal as it solidifies beyond 4 GPa at room temperature. In my previous report, I was rather hinting at Methanol:Ethanol which remains hydrostatic up to 10 GPa. In any case, the liquid PTM appears to show a higher onset T_c , a sharper drop, and no low-T tail at the foot of the transition at low pressures up to 4 GPa (c.f. Fig. 1(c), (e) and (g)). This indicates that the choice of PTM does have an appreciable effect on SC. The authors might want to include this in their manuscript.

Secondly, the new discussion of the temperature (in)dependence of the resistivity exponent α , i.e. Fig. R1, is very helpful as it does not indicate the presence of a typical quantum critical fan. This supports the authors' claim for a strange metal state. Hence, I believe it would be helpful to include this discussion at least in the SM.

Reviewer #3 (Remarks to the Author):

The authors have responded satisfactorily to my queries as a Referee in the previous submission to Nature Physics, and accordingly modified their manuscript. I think that the work in its present state warrants publication in Nature Communications.

Response to the Referees' comments and suggestions on our manuscript (NCOMMS-23-08188-T/Cai et al)

We sincerely thank the Referees for their comments and suggestions on improving our manuscript. We have seriously considered the Referees' comments and suggestions, and revised our manuscript correspondingly.

Referee #1-1: I have read carefully the revised manuscript by Cai et al., submitted to Nat. Commun, and the rebuttal letter in which the authors address the concerns and criticism of the referees of the previous review process. As far as I can tell, the replies are mostly to the point, and the authors have made sufficient changes for publishing the manuscript.

Reply: We thank the Referee's comments on our work!

Referee #1-2: However, I would like to suggest out two points which might be worth addressing beforehand: Firstly, it is certainly impressive that the authors have repeated their measurements with a liquid PTM. Unfortunately, their choice for Glycerine was probably not ideal as it solidifies beyond 4 GPa at room temperature. In my previous report, I was rather hinting at Methanol:Ethanol which remains hydrostatic up to 10 GPa. In any case, the liquid PTM appears to show a higher onset T_c , a sharper drop, and no low-T tail at the foot of the transition at low pressures up to 4 GPa (c.f. Fig. 1(c), (e) and (g)). This indicates that the choice of PTM does have an appreciable effect on SC. The authors might want to include this in their manuscript.

Reply: Following the Referee's suggestion, we have included the suggestion about effect of different pressure transmitting media (PTM) on superconductivity in the revised manuscript.

Referee #1-3: Secondly, the new discussion of the temperature (in) dependence of the resistivity exponent α , i.e. Fig. R1, is very helpful as it does not indicate the presence of a typical quantum critical fan. This supports the authors' claim for a strange metal state. Hence, I believe it would be helpful to include this discussion at least in the SM.

Reply: We appreciate the Referee's suggestion in his/her previous review report about the plot of the temperature dependence of the resistivity exponent. Following his/her suggestion, we have included these results in the SI and made a brief discussion.

Referee #3: The authors have responded satisfactorily to my queries as a Referee in the previous submission to Nature Physics, and accordingly modified their manuscript. I think that the work in its present state warrants publication in Nature Communications.

Reply: We thank the Referee's comments on our work!